# ONLINE AUCTION FOR ADS AND ORGANICS

## ABSTRACT

This paper introduces the first online blending auction mechanism design for sponsored items (ads) alongside organic items (organics), ensuring guaranteed Pareto optimality for platform revenue, advertiser utilities, and user interest (measured through clicks). We innovatively define an umbrella term, "traffic item," to encompass both organics and auctionable ad items, where an organic represents a unit of traffic to be auctioned, valued positively by attracting user interest with a fixed zero bid and payment. The online blending traffic distribution problem is thus transformed into an auction problem with unified valuation metric for the traffic item, which is subsequently formulated as an online multi-objective constrained optimization problem. We derive a Pareto equation for this optimization problem, characterizing the optimal auction mechanism set by its solution set. This solution is implemented through a novel two-stage Adaptive Modeled Mechanism Design (AMMD), which (1) trains a hypernetwork to learn a family of parameterized mechanisms, each corresponding to a specific solution of the Pareto equation, and (2) employs feedback-based online control to adaptively adjust the mechanism parameters, ensuring real-time optimality in a dynamic environment. Extensive experiments demonstrate that AMMD outperforms existing methods in both click-through rates and revenue across multiple auction scenarios, particularly highlighting its adaptability to online environments. The code has been submitted and will be released publicly.

## 1 INTRODUCTION

Online advertising has significantly contributed to the tech sector's revenue, with PwC estimating that the online ads sector will reach \$723.6 billion by 2026 (PwC, 2023). Google reported \$224 billion in advertising revenues in 2022 (Bianchi, 2023), while Meta earned \$113 billion (Dixon, 2023). The distribution of commercial advertisements (ads) is typically conducted through auctions, employing traditional techniques such as the Myerson auction (Myerson, 1981), GSP auction (Edelman et al., 2007), and VCG auction (Varian & Harris, 2014). Deep learning has emerged as a transformative force, enabling an end-to-end, modeled (i.e., parameterized) solution that learns optimal auction mechanisms directly from online traffic (Sandholm & Likhodedov, 2015). In this paradigm, an action mechanism—comprising an allocation rule and a pricing rule—is represented by one or more deep neural networks, which are trained by optimizing objectives such as revenue and regret concerning (dominant-strategy) incentive compatibility (Rahme et al., 2021). Due to their efficiency, modeled auctions have been widely adopted in industry applications (Zhang et al., 2021; Liu et al., 2021). However, in these works, organic traffic is treated as a static environment, with the auction mechanism focusing solely on ads.

This paper investigates the design of mechanisms for distributing both ads and organics. Organics play a crucial role in attracting user interest, which can lead to increased ad clicks. While they do not directly generate commercial income, they help foster a healthy ecosystem that contributes to long-term engagement. Consequently, the objectives of distributing ads and organics involve two resources: immediate commercial interest from ads and the goal of attracting user interest.

This scenario defines a new economic setting that extends beyond current theoretical understanding (as illustrated in Figure 1). Existing works are closest in two respects: (1) some use independent ranking modules to distribute personalized organics and ads (Zhao et al., 2021; Chen et al., 2022b), and (2) others unify the allocation and pricing of ads and organics, assuming a static environment (Giagkiozis & Fleming, 2015; Gunantara, 2018; Wang et al., 2021; Xia et al., 2022; Li et al., 2024b).

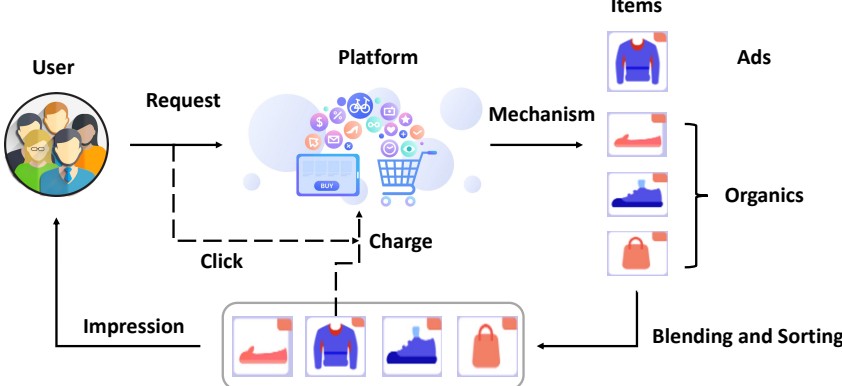

Figure 1: Mechanism design of e-commerce platforms, where only ad items generate revenue.

However, these approaches do not adequately address our setting due to their failure to consider the dynamics of traffic. In online auctions, ensuring the optimality of the mechanism necessitates that the modeled auction be adapt to online traffic characteristics, which is absent in previous studies.

To allocate and charge items with different attributes within a unified auction framework, we innovatively define a "traffic item" as an umbrella term encompassing all attribute variables. In this context, an organic is a special case, where the positive valuation arises from attracting user interest, accompanied by fixed zero bids and payments. The three objectives—platform revenue, advertiser utilities, and user interest—are abstracted into two valuation metrics (Clicks and Costs) related to the distribution and pricing of ads and organics, combined with multiple constraints. We model this challenge as an online multi-objective constrained optimization problem.

We derive a Pareto equation for this optimization problem, which characterizes the optimal auction mechanism set by its solution set. Our findings demonstrate that any independent or static blending mechanism leads to suboptimal outcomes, highlighting the need for a unified adaptive auction mechanism. The solution is implemented through an Adaptive Modeled Mechanism Design (AMMD) framework, which operates in a two-stage process.

The first stage involves training a hypernetwork to learn a family of parameterized mechanisms, each corresponding to a specific solution of the Pareto equation. The model parameters include a virtual value function for uniformly ranking both advertisements and organic items, along with incentive-compatible pricing rules to derive revenue from ads. The hypernetwork dynamically controls these mechanism parameters based on the evolving characteristics of online traffic.

The second stage utilizes feedback-based online control to adaptively adjust the mechanism parameters, ensuring real-time optimality in a dynamic environment. The optimal values of the weight parameters are influenced by traffic characteristics and the distribution of advertiser values. Since these distributions are often unknown in online auctions, we implement weight control using a feedback-based method. By continuously adjusting the mechanism parameters through the hypernetwork, we achieve an adaptive and optimal mechanism design for online scenarios.

We conduct extensive experiments demonstrating that AMMD significantly outperforms state-of-the-art methods (SOTA) across various generalized online auction scenarios. In both static and dynamic auction contexts, AMMD consistently surpasses SOTA algorithms and achieves Pareto optimality. To compare multiple objective results, such as clicks and costs, under a unified metric, we define Utopia distance as a standard measure for different mechanisms. For generalized multi-slot auctions with KPI constraints, AMMD improves the Utopia distance compared to SOTA by at least 20%, successfully achieving Pareto optimality.

## RELATED WORK

**Online modeled mechanism design.** Dütting et al. (2019) first proposed RegretNet, a deep learning-based approach to mechanism design that employs parametric models to implement al-

location and pricing rules, optimizing for revenue while adhering to incentive compatibility constraints. Shen et al. (2019) introduced MenuNet, a modeled mechanism design framework with provable optimality. By designing a misreporting agent, Rahme et al. (2021) simplified the computational complexity of incentive compatibility and improved the learning efficiency of optimal mechanisms. Given its ability to learn optimal auctions entirely from samples, modeled mechanism design has been widely adopted by e-commerce platforms, including implementations such as Deep GSP (Zhang et al., 2021) and Deep Neural Auction (DNA) (Liu et al., 2021). Recent work has incorporated more practical factors into modeled mechanism design, such as list-wise representations (Wang et al., 2022) and externality-aware ad auction design (Li et al., 2024b).

**Blending mechanisms for ads and organics.** Maintaining an appropriate percentage of ads in exposed queue has been a crucial strategy for e-commerce platforms to balance revenue and user engagement (Wang et al., 2011). Earlier research often allowed ads and organics to occupy predefined positions in blended queues. However, Yan et al. (2020) demonstrated that inserting ads into an ordered organic queue can improve allocation efficiency. Zhao et al. (2021); Xie et al. (2021) applied reinforcement learning for the insertion of ads into organic queues. Chen et al. (2022b) used a dynamic knapsack algorithm to blend ads and organics, though the ranking processes for both remained independent. Recent work by Carrion et al. (2024); Li et al. (2024b) has introduced unified virtual value functions for ranking all items. The research by Li et al. (2022; 2024a) incorporated multiple objectives and various attributes of items into the auction mechanism, providing theoretical support for the aforementioned blending mechanism design.

## 2 PROBLEM FORMULATION

When a user browses the homepage or searches for specific items, the e-commerce platform needs to allocate both organic and ad items across $k$ slots, denoted as $T_1, \cdots, T_k$. Slots are inseparable and will not display same items. The total clicks received by item $i$ allocated to slot $T_j$ is given by $click = c_i \cdot c^j$, where $c_i \sim G_i$ represents the click-through rate (CTR) of the item from bidder $i$, and $c^j$ is a constant specific to slot $T_j$. When a displayed item is clicked by a user, it has a certain probability of being ordered, thereby generating utility for the advertiser. We define the conversion value per click for bidder $i$ as $v_i \in [l_i, m_i]$, drawn from a distribution $F_i$. There are $n_1$ ad items competing for the opportunity to be displayed, while the remaining $n_2$ organic items do not participate in bidding and are not subject to charges. The widely used payment rule is Cost-Per-Click (CPC) (Qin et al., 2015), which is defined as $total\_cost = cpc \times total\_click$. Accordingly, we define the allocation of slot $j$ to item $i$ as $a_i^j$, and the CPC payment for item $i$ winning slot $j$ as $p_i^j$. We assume that the bids from these ad items are aimed at maximizing their utility:

$$U_i(a, p, v_i, b_i, c_i, \cdot) = \sum_{j=1}^{k} c_i \cdot c^j [v_i \cdot a_i^j(b_i, \cdot) - p_i^j(b_i, \cdot)]. \tag{1}$$

The allocation and payment rules are collectively referred to as the mechanism. The goal of mechanism design is to achieve Pareto optimality, which involves maximizing platform revenue, advertiser utility, and user experience. According to classic auction theory, platform revenue is defined as the total payment made by all advertisers: $R_1(a, p) = \sum_{i=1}^{n_1} \sum_{j=1}^{k} p_i^j(a, p, \cdot)$. Additionally, the objective of maximizing advertiser utility is typically simplified to ensuring incentive compatibility (IC) and individual rationality (IR) constraints. User engagement is usually measured by the total clicks on all displayed items, represented as $R_2(a, p) = \sum_{i=1}^{n_1+n_2} \sum_{j=1}^{k} c_i \cdot c^j \cdot a_i^j$. Previous research has indicated that both excessively high and low percentages of ad exposure (PAE) can be detrimental to the platform's overall health. Therefore, maintaining a specific PAE is often imposed as a constraint in mechanism optimization (Zhang et al., 2018; Liao et al., 2022).

**Definition 1** (Multi-slot auction as a constrained multi-objective optimization problem). *The mechanism design can be defined as a constrained optimization problem:*

$$max_{(a,p)} \ \Phi[R_1(a, p), R_2(a, p)] \quad s.t. \ \forall b_i \neq v_i, \ U_i(a, p, b_i, \cdot) \leq U_i(a, p, v_i, \cdot) \ (IC);$$

$$\forall v_i, \ U_i(a, p, v_i, \cdot) \geq 0 \ (IR); \quad \sum_{i=1}^{n_1} \sum_{j=1}^{k} a_i^j = k \cdot \lambda_0 \ (PAE \ constraint) \tag{2}$$

Traditional auctions that aim to maximize total payment fail to meet the requirements in the above setting. Therefore, we define the constrained multi-objective multi-slot auction in definition 1.

In Definition 1, the position CTR $c^j$ is typically assumed to decrease with increasing position index $j$ (Cavallo et al., 2018). The function $\Phi$ represents a combination of multiple objectives in the mechanism. Incentive compatibility (IC) and individual rationality (IR) ensure that advertisers' truthful bidding strategy, where bids equal their valuations $(b = v)$, maximizes their utility. The parameter $\lambda_0$ represents the optimal PAE. This definition is designed for scalability, allowing for the inclusion of various key performance indicator (KPI) constraints in addition to the PAE, such as expected conversion rate (CVR), diversity, and return on investment (ROI).

## 3   A UNIFIED MECHANISM DESIGN FRAMEWORK WITH ADS AND ORGANICS

We first consider the optimal mechanism design for a single slot without KPI constrains. We assume that the value and CTR of bidder $i \in \{1, \cdots, n_1\}$ are $v_i \sim F_i$ and $c_i \sim G_i$, respectively. The joint distributions are $F = F_1 \times \cdots \times F_{n_1}$ and $G = G_1 \times \cdots \times G_{n_1}$, with density functions $f$ and $g$. There are also $n_2$ organic items with CTR $c_j^o \sim G_j^o$. First, we show that for the multi-objective optimization problem defined in Equation 2, adopting an arbitrary combination function $\Phi$ will result in the final optimization result being contained within the Pareto region defined below.

**Definition 2** (Pareto Region). *Given $x \in X$ and functions $Q_1(x)$, $Q_2(x)$, $D(x)$ is defined as:*

$$\forall x_1 \in D(x), x_2 \in X \setminus \{x_1\}, \text{ if } Q_1(x_1) \neq Q_1(x_2) \text{ or } Q_2(x_1) \neq Q_2(x_2),$$
$$\text{we have } Q_1(x_1) > Q_1(x_2) \text{ or } Q_2(x_1) > Q_2(x_2) \tag{3}$$

**Theorem 1.** *Given $Q_1(x)$, $Q_2(x)$ and $\Phi(Q_1, Q_2)$, which satisfies that $\forall i \in \{1, 2\}, \nabla_{Q_i}\Phi > 0$. If $x = argmax\ \Phi(Q_1, Q_2)$, then we have $x \in D(x)$, which is the Pareto region.*

Theorem 1 demonstrates that employing a complex combination function $\Phi$ does not improve one objective function without diminishing another. Therefore, we define the objective of mechanism as simple linear function $cost + \alpha click = R_1 + \alpha R_2$. When the slot is sold to bidder $i$, the platform will receive multi-objective revenue $c_i \cdot p_i(\vec{v}, \vec{c}, \vec{c^o}) + \alpha c_i$. If the slot is not sold as an ad, it will be exposed as an organic. Therefore, we denote the revenue for not selling the slot as $\alpha c_0 = \alpha \max(\vec{c}^o)$.

The seller needs to select allocation and payment rule $(a, p)$ to maximize multi-objective revenue:

$$R_1(a, p) + \alpha R_2(a, p) = \int_G \int_F \int_{G^o} [\alpha c_0 (1 - \sum_{i=1}^{n_1} a_i(\vec{v}, \vec{c}, c_0))$$

$$+ \sum_{i=1}^{n_1} (c_i \cdot p_i(\vec{v}, \vec{c}, c_0) + \alpha a_i(\vec{v}, \vec{c}, c_0) c_i)]\ f(\vec{v}) g(\vec{c}) g^o(c_0) dv dc dc_0 \tag{4}$$

Similar to Myerson mechanism (Myerson, 1981), we use feasible to denote the mechanism both satisfying IC and IR constrains. Then we have the following conclusion:

**Theorem 2.** *If the mechanism $(a, p)$ is feasible, maximizing Equation 4 is equivalent to maximizing*

$$\int_G \int_F \int_{G^o} \sum_{i=1}^{n_1} a_i(\vec{v}, \vec{c}, \vec{c}^o) [c_i(v_i - \frac{1 - F_i(v_i)}{f_i(v_i)}) + \alpha(c_i - c_0)] df\, dg\, dg^o \tag{5}$$

*where we use $df, dg, dg^o$ to denote $f(v)dv, g(c)dc, g^o(c_0)dc_0$.*

This theorem indicates that the revenue of feasible mechanisms is solely related to the allocation rule, and the slot should be allocated to the advertiser with highest

$$c_i(v_i - \frac{1 - F_i(v_i)}{f_i(v_i)}) + \alpha(c_i - c_0), \text{ if } \max_i c_i(v_i - \frac{1 - F_i(v_i)}{f_i(v_i)}) + \alpha(c_i - c_0) \geq 0 \tag{6}$$

otherwise the slot should be allocated to the organic item with CTR $c_0$.

To build a unified mechanism framework, we define the value of organics is zero $v_i^o = 0, \forall i \in \{Org\}$ and thus every item including ads and organics can be represented as a "traffic item" $(v_i^*, c_i^*)$. And we define

$$\Psi(v_i^*) = 0, \text{ if } v_i^* \equiv 0, \text{ otherwise } v_i^* - \frac{1 - F_i(v_i^*)}{f_i(v_i^*)} \tag{7}$$

We denote the distribution $F_i$ as a normal distribution if $\Psi(v_i^*)$ is an increasing function of $v_i^*$ (This is a generally adopted assumption in mechanism design which comes from (Myerson, 1981)). Then we define the unified auction for ads and organics:

**Definition 3** (Unified auction for ads and organics). *Given all ads and organics $(v_i^*, c_i^*)$ ($v_i^* \in \{0\} = V_i$ if $i \in \{Org\}$, otherwise $v_i^* \in [l_i, m_i] = V_i$), and we assume that all the value distributions $F_i$ are normal. The allocation rule and payment rule are as follows:*

- *The slot is allocated to item $i$, where $i = argmax_i[(c_i^* \Psi(v_i^*) + \alpha c_i^*) \cdot \mathbf{I}(v_i^* \in V_i)]$.*

- *The cost per click $p_i(v_i^*, c_i^*)$ is $max[\Psi^{-1}((c_j^* \Psi(v_j^*) + \alpha c_j^* - \alpha c_i^*)/c_i^*), min(v^* \mid v^* \in V_i)]$, where $j = argmax_{j \neq i} c_j^* \Psi(v_j^*) + \alpha c_j^*$.*

Here $\mathbf{I}$ is a characteristic function to prevent bids lower than $l_i$, its value is 1 when $v_i^* \in V_i$, otherwise 0. Similar to the Myerson mechanism, $c_i^* \Psi(v_i^*) + \alpha c_i^*$ can be understood as the virtual value function in this definition. And we have the following theorem for this unified auction.

**Theorem 3.** *The auction defined in Definition 3 has the following properties.*

- *It satisfies IC and IR for all the items, $\forall i$, $v_i^* \geq p_i(v_i^*, c_i^*) \geq 0$.*

- *It maximizes the multi objective $R_1(a, p) + \alpha R_2(a, p)$, which is the same to Equation 5.*

- *The percentage of ads satisfies $\mathbf{E}\lambda_{ad} = \lambda_0$ if and only if*

$$P_{\vec{v} \sim F, \vec{c} \sim G}(max_{i \in \{Ad\}} c_i \Psi(v_i) + \alpha(c_i - c_0) > 0) = \lambda_0. \tag{8}$$

We denote Equation 8 as the Pareto equation, which shows that changing the multi-objective weight $\alpha$ can control the proportion of ads in the impression. To achieve Pareto optimality while satisfying PAE constraint, we can solve for the corresponding $\alpha$ for a given $\lambda_0$ and any distribution $F, G, G^o$ using the Pareto equation, and then execute the mechanism defined in Definition 3.

In previous works, exposed items were usually charged within the ads queue and blended with the organics queue in a fixed proportion without changing the order (Ouyang et al., 2020; Li et al., 2020). The approximate optimal revenue of these static blending mechanisms is

$$\alpha(1 - \lambda_0)R_2^{org} + \lambda_0[R_1^{ad}(a, p) + \alpha R_2^{ad}(a, p)] \tag{9}$$

Compared to the Equation 5, we have the following property:

**Theorem 4.** *Given the weight $\alpha$ and corresponding $\lambda_0 \in (0, 1)$ which satisfies Definition 3, we have the following conclusion:*

$$max_{a,p} R_1(a, p) + \alpha R_2(a, p) > max_{a,p}(1 - \lambda_0)\alpha R_2^{org} + \lambda_0[R_1^{ad}(a, p) + \alpha R_2^{ad}(a, p)] \tag{10}$$

*where the left part comes from Equation 5 and right part comes from Equation 9.*

This indicates that uniformly ranking and charging ads and organics will promote revenue.

**Remark 1.** *Compared to the traditional optimal auction theory (revenue maximization as a single objective), this framework allows for arbitrary extensions for item attributes. We provide an example of the CVR attribute and the corresponding Pareto Equation in the Supplementary Material.*

**Remark 2.** *All the proofs are given in detail in the Supplementary Material. The main idea of the proof is to establish the relationship between the multi-objective revenue of the mechanism and the advertiser's utility, thereby simplifying formula 4.*

## 4 ADAPTIVE MODELED MECHANISM DESIGN FOR ONLINE AUCTIONS

In this section, we propose the Adaptive Modeled Mechanism Design (AMMD), illustrated in Figure 2. AMMD adopts the virtual value similar to Equation 6 and ensures incentive compatibility through Vickrey-Clarke-Groves (VCG) pricing rules. It leverages a hypernetwork to learn the optimal mechanism parameters corresponding to different objective weights. To adapt the mechanism to changing traffic characteristics in online auctions, we use an online control algorithm to dynamically adjust the multi-objective weights. By leveraging the hypernetwork to update the model parameters, AMMD achieves Pareto optimality in constrained online environments. We take the CVR constrains as an example to illustrate that AMMD has good scalability for arbitrary objectives.

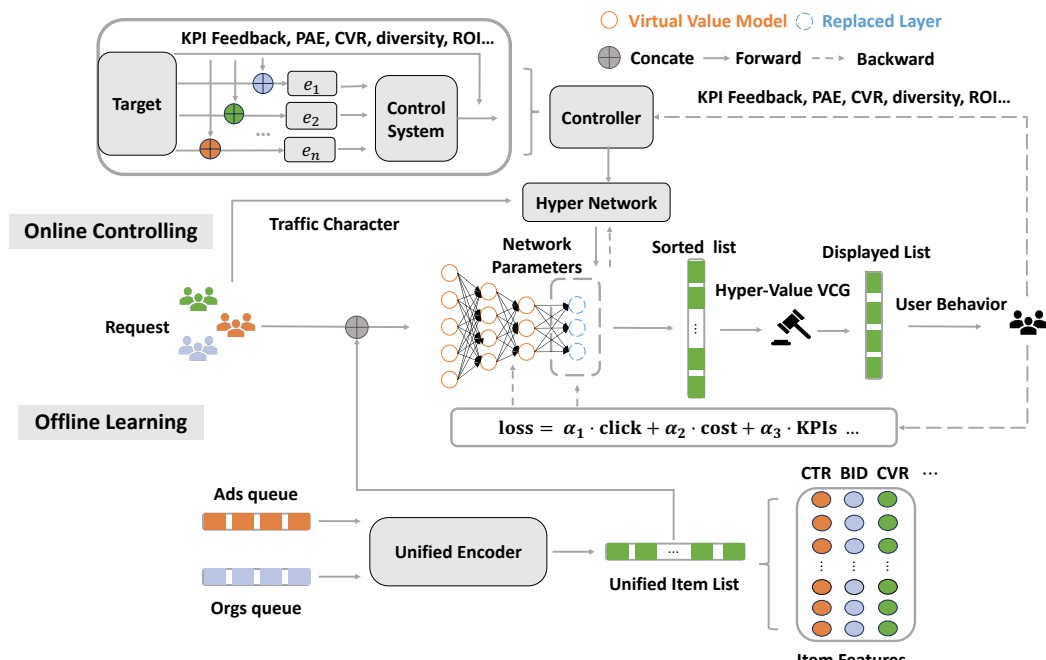

Figure 2: The AMMD framework consists of two main components: offline learning and online controlling. In the offline learning stage, both the virtual value model and the hypernetwork parameters are updated simultaneously. During the online controlling stage, the weights produced by the controller adjust the parameters of the replaced layers through the hypernetwork, allowing the mechanism to adapt to evolving traffic characteristics in real time.

## 4.1 MODELED VIRTUAL VALUE AND VCG PRICING

From Theorem 2, we observe that once the sorting rules are fixed, the pricing rules that ensure incentive compatibility are unique. Consequently, we can optimize the sorting process within the family of IC mechanisms to achieve multi-objective Pareto optimality. Thus, we decompose the optimal mechanism in a static setting into two components: the ranking rule based on the modeled virtual value function and the VCG pricing rule, which is proven to satisfy incentive compatibility.

For online dynamic environments, we replace the virtual value function in Equation 6 with a parameterized model trained using deep learning. Let $M(\cdot, \theta)$ represent this modeled virtual value function. The modeled score for each "traffic item" is calculated using its attributes, including the values $(\vec{v}, 0)$ and the CTR $(\vec{c}, \vec{c}^o)$, CVR $(\vec{z}, \vec{z}^o)$: $q_i(v_i, c_i, z_i) = M[(v_i, c_i, z_i), \theta]$.

In order to maximize the mutil-objective revenue, the allocation rule should be:

$$a_i^j(v_i, c_i, z_i) = \operatorname{argmax} \sum_{j=1}^{k} q_i(v_i, c_i, z_i) \cdot c^j \cdot a_i^j(v_i, c_i, z_i) \tag{11}$$

In real e-commerce scenarios, the simultaneous impression of ad and organic items can influence each other's value and CTR. This phenomenon is known as externality, meaning that $c_i = c_i(a^1, \cdots, a^k)$ and $v_i = v_i(a^1, \cdots, a^k)$. Recent studies have investigated optimal ranking methods in the presence of externalities and have improved the efficiency of slot allocation Chen et al. (2022a); Li et al. (2023). However, in this work, we focus on constrained multi-objective optimal mechanisms and therefore assume the absence of externalities. Algorithms that address externalities can be directly integrated into our framework, and this will be considered in future research.

In multi-slot auctions, VCG pricing has been proven to satisfy incentive compatibility constraints (Varian & Harris, 2014). It can be understood as the payment equating to the loss in social welfare imposed on other items, where social welfare is defined as the sum of virtual values. We denote the

inverse function $q^{-1}$ for deriving $v$ with fixed $q, c, z$. Then the VCG pricing rule can be written as:

$$p_i^j(v_i, c_i, z_i) = q_i^{-1}[[\max_{s_1, \cdots, s_k \in \{1, \cdots, N\} \setminus \{i\}} \sum_{m=1}^{k} q_{s_m}(v_{s_m}, c_{s_m}, z_{s_m}) \cdot c^m \cdot a_{s_m}^m(v_{s_m}, c_{s_m}, z_{s_m})$$

$$- \max_{s_1, \cdots, s_k \in \{1, \cdots, N\} \setminus \{i\}} \sum_{m \neq j}^{k} q_{s_m}(v_{s_m}, c_{s_m}, z_{s_m}) \cdot c^m \cdot a_{s_m}^m(v_{s_m}, c_{s_m}, z_{s_m})]/c^m]$$

(12)

It is worth noting that this charging rule may result in charging below the lower bound of the value distribution $V_i$ (sometimes leads to negative payments). To avoid this problem, we actually adopt a payment rule similar to that in Definition 3: $p = \max[p_i^j(v_i, c_i, z_i), \min(v^* \mid v^* \in V_i)]$. The above are the modeled ranking and pricing rules in AMMD. We will introduce the training and online execution of the parameterized model in detail in the next section.

## 4.2 Hypernetwork for Adaptive Modeled Mechanism Design

For a static environment, we can directly adopt the loss function to train mechanism parameter $\theta$:

$$Loss(\theta) = -R_1(\theta) - \alpha R_2(\theta) + \beta R_3(\theta) = -\sum_{m=1}^{k} \sum_{i=1}^{N} c^m \cdot c_i \cdot p_i^m(\theta)$$

$$- \alpha \sum_{m=1}^{k} \sum_{i=1}^{N} c^m \cdot c_i \cdot a_i^m(\theta) + \beta \text{Relu}(\rho_0 - \sum_{m=1}^{k} \sum_{i=1}^{N} c^m \cdot c_i \cdot a_i^m(\theta) \cdot z_i^m)$$

(13)

However, online auctions need to deal with dynamically changing traffic characteristics and bidding environments. Using modeled virtual value $\theta$ trained with static optimal weights $(\alpha, \beta)$ cannot maintain optimality. To adapt the mechanism parameters to the online environment, we introduce a hypernetwork module. We divide the parameter of the mechanism $\theta$ into a fixed part $\theta_{-w}$ and a controllable part $\theta_w$, which is generated by the hypernetwork as $\theta_w = H(\vec{w}, \theta_H)$. The input to this hypernetwork $\vec{w}$ consists of multi-objective weights $(\alpha, \beta)$ and real-time bidding information $F, C, Z$. The output is the parameters of the replaced layer, as shown in Figure 2.

To train the hypernetwork, we randomly sample a set of weight parameters $(\alpha, \beta)$ and value distributions as inputs during each iteration. By applying the same parameters to weight the loss function, we simultaneously train both the hypernetwork parameters $\theta_H$ and the fixed mechanism model parameters $\theta_{-w}$ through backpropagation (as detailed in Supplementary Metarial Algorithm 1). Since the modeled mechanism involves multiple sorting operations, we utilize the differentiable sorting operator proposed in Grover et al. (2019) to support the training process.

By training the hypernetwork, we develop a family of mechanism parameters with strong generalization capabilities. As the weight parameters and bidding environment evolve, the modeled auction can seamlessly adapt to the instantaneously optimal configuration.

## 4.3 Online Control for Adaptive Weights Selection

According to the Pareto Equation, the multi-objective weights $\vec{w}$ that satisfy the constraints in the optimal mechanism family are necessary. However, in online environments where traffic characteristics fluctuate in real time, it is impractical to precisely determine the optimal weights.

Assuming a higher expected CTR for organic items, it is evident that the weight parameter $\alpha$ is positively correlated with the proportion of organic items. We design an online control system that continuously compares the real-time proportion of organic items to a predefined target proportion $\lambda_{target}$. At each time step, the actual proportion of organic items is compared to the target, and the difference is recorded as the error $e(t) = \lambda_{target}^t - \lambda_{org}^t$.

We employ a Proportional-Integral-Derivative (PID) controller, which is an efficient algorithm in online feedback control (Yang et al., 2019; Balseiro et al., 2022). The control signal generated by the PID controller consists of proportional, integral, and derivative terms, which adjust the weight

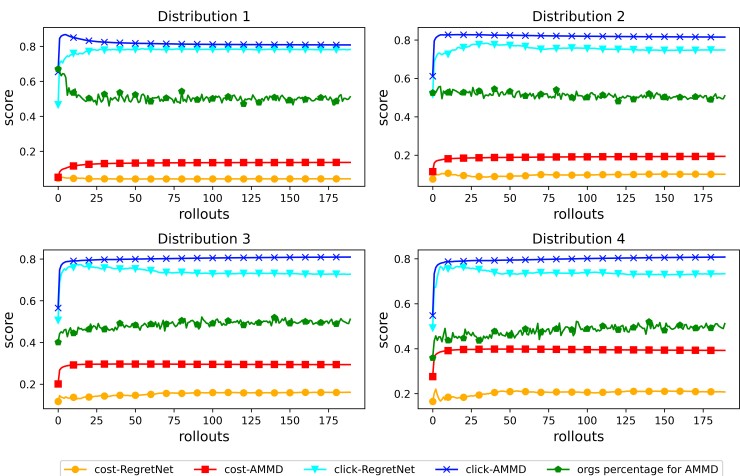

Figure 3: Experiments in independent identical multi-slot auctions with various value distributions.

parameters within a reasonable range using an exponential function.

$$\alpha^{t+1} = \alpha^t \cdot e^{u(t)} \quad u(t) = k_p e(t) + k_i \sum_{k=t-l}^{t-1} e(k) + k_d(e(t) - e(t-1)) \tag{14}$$

Here $(k_p, k_i, k_d)$ and $l$ are hyper parameters for the algorithm. We can similarly construct PID controllers for the other weight parameters. According to the research of Zhao & Guo (2017); Zhao & Yuan (2024), the PID method demonstrates strong convergence in both static environment control and dynamic environment tracking, which provides theoretical support for the application of AMMD in online auction scenarios. The full AMMD algorithm is detailed in Supplementary Material 2.

## 5 EXPERIMENTS

### 5.1 EXPERIMENTS IN INDEPENDENT IDENTICAL MULTI-SLOT AUCTIONS

We first conduct experiments in independent identical multi-slot auction scenarios. In this environment, the CTR is determined solely by the attributes of the allocated items. Since the auctions between different slots are independent, this setting can be simplified to repeated single-slot auctions. Our comparison baseline is the static modeled mechanism trained with RegretNet (Dütting et al., 2019). RegretNet is a significant work in modeled mechanism design that efficiently achieves optimal auctions through deep learning. However, since RegretNet only supports fixed multi-objective weights, we trained multiple groups simultaneously with fixed objective weights. To maintain the PAE for static RegretNet mechanisms, the probability of displaying organic item is set to $\lambda_0 = 50\%$.

**Implementation:** Experiments in independent identical multi-slot auctions consists of both static and dynamic scenarios. In static scenarios, We show that the AMMD framework enables mechanism to simultaneously learn the optimal auction under any traffic distribution. We randomly generate $n_1 = 2$ ads with different value distribution $F_{ex} \sim U[0, 0.5 \times ex], ex = 1, \cdots, 4$ and fixed CTR distribution $C^{ad} \sim U[0, 1]$. There are also $n_2 = 2$ organic items with expected maximum CTR $c_0 \sim U[0, 2]$. We train and test AMMD using samples from different value distribution $F_{ex}$. Auctions in all scenarios utilize a single AMMD model with independent PID-controlled weights.

We also simulate online auction environments, where traffic characteristics vary periodically over time. Our experiments consist of mechanisms that can sense the real-time distribution (online) and those that can only access expected distribution (offline). In this experiment, we plot the Pareto curves learned by RegretNet with different weights $\alpha \in (0.1, 2)$.

**Results:** The static experimental results are shown in Figure 3. As observed, the mechanism trained by AMMD Pareto-dominates the optimal mechanism learned by RegretNet, highlighting the

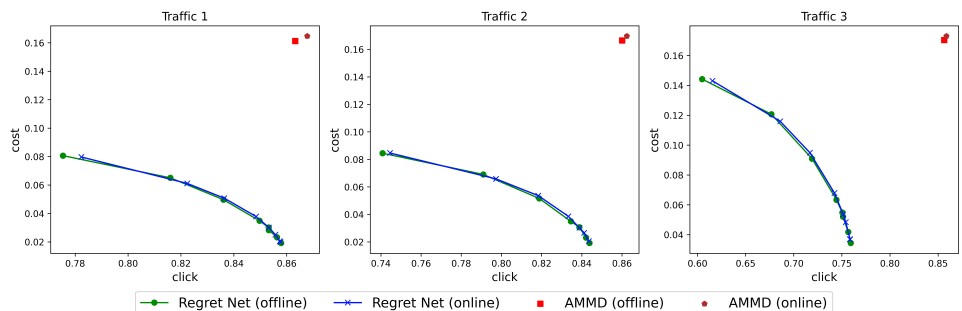

Figure 4: Comparison between AMMD and Pareto curve of RegretNet in dynamic environments.

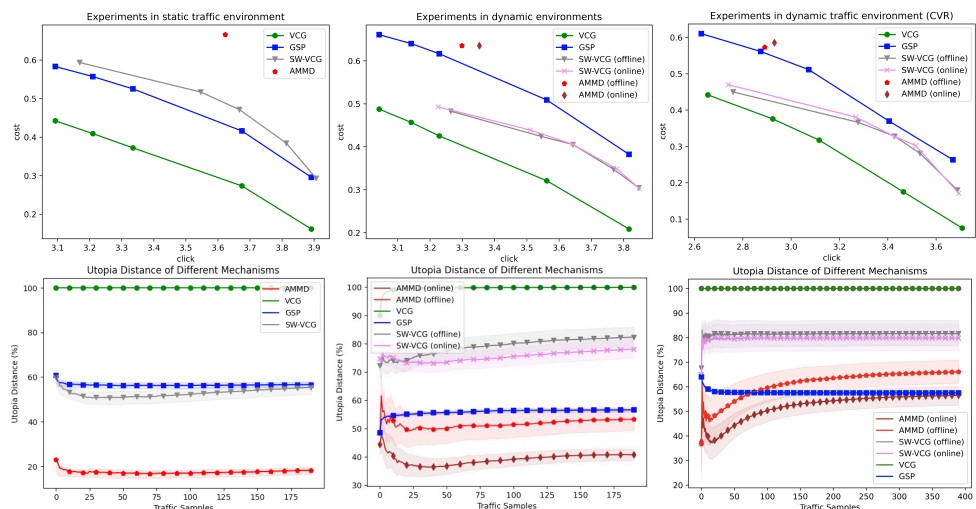

Figure 5: Comparison between AMMD and Pareto curve of GSP, VCG and SW-VCG in static (left), dynamic (middle) and CVR constrained (right) scenarios.

advantages of an adaptive mechanism and supporting Theorem 4. Experiment details is given in the Supplementary Material. The AMMD algorithm effectively maintains the PAE close to $50\%$, with fluctuations not exceeding $3\%$. These fluctuations arise from the varying quality of ads and organic items in batches of data sourced from random sampling, leading to instability in their virtual value differences. When the average quality of ads in an episode is higher, it is reasonable to allocate more slots to them. This also demonstrates that the application of online controller allows AMMD to achieve more efficient allocation in each auction while ensuring the PAE constraint.

Figure 4 depicts the mechanisms learned by RegretNet with different multi-objective weights alongside our AMMD mechanism in dynamic experiments. As shown, RegretNet with varying weights forms a Pareto curve, representing a family of optimal static mechanisms. Notably, AMMD with adaptive weights significantly surpasses the Pareto curve of the static RegretNet mechanism. This result demonstrates that AMMD maximizes multi-objective revenue by enhancing the allocation efficiency of each auction while maintaining incentive compatibility and proportion constraints.

### 5.2 EXPERIMENTS IN GENERALIZED MULTI-SLOT AUCTIONS

**Implementation:** We simulate an environment similar to online multi-slot auctions. We randomly generated multiple ads and organic items for auction across four slots, with base CTRs of $(1.0, 0.8, 0.6, 0.5)$. In addition to static and dynamic scenarios similar to independent identical multi-slot auctions, the experimental setting includes experiments with CVR constraints to illustrate the adaptability of AMMD to extended constraints. Our comparison methods consist of the VCG and GSP mechanisms, as well as the modeled mechanism Score-Weighted VCG (SW-VCG) (Li et al., 2024b). For the VCG and GSP mechanisms, all items are sorted according to their virtual

| Mechanism ($\alpha$, $\beta$) | click | cost | PAE | CVR | IC | Utopia distance |
|---|---|---|---|---|---|---|
| VCG (0.01, 0.6) | 2.635 | 0.430 | 50% | 0.103 | ✓ | 100% |
| VCG (0.5, 5.5) | 3.432 | 0.187 | 50% | 0.103 | ✓ | |
| GSP (0.01, 0.6) | 2.601 | 0.604 | 50% | 0.103 | ✗ | 59.7% |
| GSP (0.5, 5.5) | 3.370 | 0.378 | 50% | 0.103 | ✗ | |
| offline SW-VCG (0.01, 0.4) | 2.763 | 0.441 | 50% | 0.103 | ✓ | 82.3% |
| offline SW-VCG (0.5, 4.5) | 3.570 | 0.235 | 50% | 0.102 | ✓ | |
| online SW-VCG (0.01, 0.4) | 2.744 | 0.461 | 50% | 0.103 | ✓ | 81.4% |
| online SW-VCG (0.5, 4.5) | 3.533 | 0.277 | 50% | 0.101 | ✓ | |
| offline AMMD | 2.932 | 0.571 | $49.6\% \pm 9.3\%$ | 0.103 | ✓ | 64.7% |
| online AMMD | 2.954 | 0.588 | $49.7\% \pm 3.2\%$ | 0.103 | ✓ | 59.2% |

Table 1: Experiments detail in multi-slots auctions

values, and the cost per click is calculated using the corresponding pricing rules. It is important to note that the GSP rule is non-incentive compatible. For the modeled SW-VCG mechanism, we first train a modeled ranking network and then apply the VCG pricing rule. All the above mechanisms use a family of static weights (detailed in Supplementary Material) to simulate Pareto curve.

In order to compare all algorithms under a unified metric, we refer to the study of Carrion et al. (2024) and define the Utopia distance (detailed in Supplementary Material). For the objectives in our tests (including clicks, costs, and CVR), we define the maximum value of the results obtained by all algorithms as the Utopia point. The minimum distance between each algorithm and the Utopia point is defined as the Utopia distance. The Utopia distance of VCG is set to 100% for comparison.

**Results:** We compare the performance of mechanisms in both static and dynamic environments, with the results shown in Figure 5 (above). As can be seen from the figures, AMMD achieves Pareto optimality in all experiment settings. In the CVR-constrained experiment (requiring a CVR $\geq 0.1$), the increase in the Utopia distance of the AMMD algorithm over time reflects the PID control process satisfying the constraints. Initially, both CVR and the proportion of organic items were below the target values, resulting in certain losses in clicks and costs during the feedback adjustment process. In the convergence stage, the AMMD algorithm improves the Utopia distance by at least 20% in all settings (in Table 1). The reason why GSP achieved good results is that it seriously violated IC constraints (analysed in Supplementary Material).

**Remark 3.** *We verify that AMMD satisfies incentive compatibility in the supplementary material.*

**Remark 4.** *As part of an ablation study, we can observe that the improvement of the AMMD algorithm in the 'online' group trained with online data is more significant. This is due to the static weighted mechanism's inability to adjust the display proportion of ad items based on ad quality, leading to no revenue improvement even when utilizing real-time distribution information. In contrast, AMMD with adaptive weights significantly surpasses the Pareto curve of static mechanisms, demonstrating effective utilization of online information.*

## 6 CONCLUSION

In this work, we propose Adaptive Modeled Mechanism Design (AMMD) for multi-slot auctions that incorporate both ads and organics. We define the ranking and pricing rules based on a virtual value function for multi-objective auctions, demonstrating their effectiveness in maximizing revenue while maintaining incentive compatibility. We establish the Pareto equation that links multi-objective weights and constraints, enabling the decomposition of the constrained multi-objective optimization problem in online auctions into offline learning of a static mechanism family and online weight control. In AMMD, a hypernetwork is employed to learn a family of optimal static mechanisms, each tailored to specific traffic characteristics. Additionally, PID controllers are used to update the weight parameters online, ensuring the mechanism's optimality as traffic distribution evolve. We validate the effectiveness of AMMD through a series of multi-slot auction experiments. The results indicate that AMMD achieves Pareto optimal mechanisms across all tested environments. We anticipate that this work will encourage further research on multi-objective auctions that blend ads and organic content on e-commerce platforms.

## REPRODUCIBILITY STATEMENT

The following information can be found in the appendix of the paper and the submitted code. Proofs of all novel claims and theorems (see appendix A.1). A conceptual outline and pseudocode description of AI methods introduced (see appendix A.2). Any code required for pre-processing data and for conducting and analyzing the experiments is included (see the submitted code). The experiment details, including the computing infrastructure used for running experiments (hardware and software), including GPU/CPU models; amount of memory; operating system; names and versions of relevant software libraries and frameworks; all final (hyper-)parameters used for each model/algorithm in the paper's experiments (see appendix A.3).

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

# A APPENDIX

## A.1 OMITTED PROOFS

**Theorem 1.** *Given $Q_1(x)$, $Q_2(x)$ and $\Phi(Q_1, Q_2)$, which satisfies that $\forall i \in \{1, 2\}, \nabla_{Q_i}\Phi > 0$. If $x = argmax\ \Phi(Q_1, Q_2)$, then we have $x \in D(x)$, which is the Pareto region.*

*Proof.* Assuming that:

$$x = \operatorname{argmax}\ \Phi_1(Q_1, Q_2) \tag{15}$$

If $x \notin D(x)$, then there must exists $x'$,

$$Q_1(x) \le Q_1(x'),\ Q_2(x) < Q_2(x')\ or\ Q_1(x) < Q_1(x'),\ Q_2(x) \le Q_2(x') \tag{16}$$

Since $\forall i \in \{1, 2\}, \nabla_{Q_i}\Phi > 0$, we have:

$$\Phi[Q_1(x), Q_2(x)] < \Phi[Q_1(x'), Q_2(x')] \tag{17}$$

This is conflict with $x = \operatorname{argmax}\ \Phi(Q_1, Q_2)$. Therefore, Theorem 1 holds. $\qquad\square$

**Theorem 2.** *If the mechanism $(a, p)$ is feasible, maximizing Equation 4 is equivalent to maximizing*

$$\int_G \int_F \int_{G^o} \sum_{i=1}^{n_1} a_i(\vec{v}, \vec{c}, \vec{c}^o)[c_i(v_i - \frac{1 - F_i(v_i)}{f_i(v_i)}) + \alpha(c_i - c_0)] df\,dg\,dg^o \tag{18}$$

*where we use $df, dg, dg^o$ to denote $f(v)dv, g(c)dc, g^o(c_0)dc_0$.*

*Proof.* We denote the value of all bidders as $\vec{v} = (v_i, v_{-i})$, the click-through rate of all bidders as $\vec{c} = (c_i, c_{-i})$. The utility of bidder $i$ with value $v_i \in [l_i, m_i]$ is :

$$U_i(a, p, v_i) = \int_{G^o} \int_{G_i} c_i \int_{F_{-i}} \int_{G_{-i}} v_i \cdot a_i(v_i, v_{-i}, c_i, c_{-i}, c_0) - p_i(v_i, v_{-i}, c_i, c_{-i}, c_0) df_{-i} dg dg^o \tag{19}$$

The multi-objective revenue of the seller is:

$$R_1(a, p) + \alpha R_2(a, p)$$

$$= \alpha \int_G \int_F \int_{G^o} c_o[1 - \sum_{i=1}^{n_1} a_i(\vec{v}, \vec{c}, c_0)] + \sum_{i=1}^{n_1} [c_i \cdot p_i(\vec{v}, \vec{c}, c_0) + \alpha a_i(\vec{v}, \vec{c}, c_0) \cdot c_i] df\,dg\,dg^o$$

$$= \int_G \int_F \int_{G^o} \sum_{i=1}^{n_1} [c_i p_i(\vec{v}, \vec{c}, c_0) - c_i a_i(\vec{v}, \vec{c}, c_0) v_i] df\,dg\,dg^o + \int_G \int_F \int_{G^o} \alpha c_0 [1 - \sum_{i=1}^{n_1} a_i(\vec{v}, \vec{c}, c_0)]$$

$$+ \sum_{i=1}^{n_1} [\alpha \cdot c_i a_i(\vec{v}, \vec{c}, c_0) + c_i a_i(\vec{v}, \vec{c}, c_0) v_i] df\,dg\,dg^o \tag{20}$$

We record the two part above as :

$$I_1 = \int_G \int_F \int_{G^o} \sum_{i=1}^{n_1} [c_i p_i(\vec{v}, \vec{c}, c_0) - c_i a_i(\vec{v}, \vec{c}, c_0) v_i] df \, dg \, dg^o$$

$$I_2 = \int_G \int_F \int_{G^o} \alpha c_0 [1 - \sum_{i=1}^{n_1} a_i(\vec{v}, \vec{c}, c_0)] + \sum_{i=1}^{n_1} [\alpha \cdot c_i a_i(\vec{v}, \vec{c}, c_0) + c_i a_i(\vec{v}, \vec{c}, c_0) v_i] df \, dg \, dg^o \tag{21}$$

The IC constrains indicates that:

$$\forall s_i \in [l_i, m_i], U_i(a, p, s_i) \le U_i(a, p, v_i) \tag{22}$$

which means that:

$$U_i(a, p, v_i) \ge \int_G \int_{F_{-i}} \int_{G^o} c_i [v_i \cdot a_i(s_i, v_{-i}, c_i, c_{-i}, c_0) - p_i(s_i, v_{-i}, c_i, c_{-i}, c_0)] dg \, df_{-i} \, dg^o$$

$$= \int_G \int_{F_{-i}} \int_{G^o} c_i s_i a_i(s_i, v_{-i}, c_i, c_{-i}, c_0) - c_i p_i(s_i, v_{-i}, c_i, c_{-i}, c_0)$$

$$+ (v_i - s_i) c_i a_i(s_i, v_{-i}, c_i, c_{-i}, c_0) dg \, df_{-i} \, dg^o \tag{23}$$

$$= U_i(a, p, s_i) + (v_i - s_i) \int_G \int_{F_{-i}} \int_{G^o} c_i a_i(s_i, v_{-i}, c_i, c_{-i}, c_0) dg \, df_{-i} \, dg^o$$

We denote:

$$Q_i(a, v_i) = \int_G \int_{F_{-i}} \int_{G^o} c_i a_i(v_i, v_{-i}, c_i, c_{-i}, c_0) dg \, df_{-i} \, dg^o \tag{24}$$

Using equation (23) twice, we get:

$$\forall t_i, \ (t_i - s_i) Q_i(a, s_i) \le U_i(a, p, t_i) - U_i(a, p, s_i) \le (t_i - s_i) Q_i(a, t_i) \tag{25}$$

From equation (25), we can see that $Q_i(a, s_i)$ is increasing in $s_i$. This inequalities can be written for any $\delta > 0$:

$$Q_i(a, s_i) \delta \le U_i(a, p, s_i + \delta) - U_i(a, p, s_i) \le Q_i(a, s_i + \delta) \delta \tag{26}$$

Since $Q_i(a, s_i)$ is increasing in $s_i$, it is Riemann integrable. Then we have the following property:

$$\int_{l_i}^{t_i} Q_i(a, s_i) ds_i = U_i(a, p, t_i) - U_i(a, p, l_i) \tag{27}$$

According to this, we have:

$$I_1 = \sum_{i=1}^{n_1} \int_G \int_{G^o} \int_{F_{-i}} \int_{F_i} [c_i p_i(\vec{v}, \vec{c}, c_0) - c_i a_i(\vec{v}, \vec{c}, c_0) v_i] df_i \, df_{-i} \, dg \, dg^o$$

$$= -\sum_{i=1}^{n_1} \int_G \int_{G^o} \int_{F_{-i}} \int_{l_i}^{m_i} [U_i(a, p, l_i) + \int_{l_i}^{t_i} Q_i(a, s_i) ds_i] f_i(t_i) dt_i \, df_{-i} \, dg \, dg^o$$

$$= -\sum_{i=1}^{n_1} \int_G \int_{G^o} \int_{F_{-i}} [U_i(a, p, l_i) + \int_{l_i}^{m_i} \int_{s_i}^{m_i} f_i(t_i) Q_i(a, s_i)] dt_i \, ds_i \, df_{-i} \, dg \, dg^o \tag{28}$$

$$= -\sum_{i=1}^{n_1} \int_G \int_{G^o} \int_{F_{-i}} [U_i(a, p, l_i) + \int_{l_i}^{m_i} (1 - F_i(s_i)) Q_i(a, s_i)] dt_i \, ds_i \, df_{-i} \, dg \, dg^o$$

$$= -\sum_{i=1}^{n_1} \int_G \int_{G^o} \int_{F_{-i}} [U_i(a, p, l_i) + \int_{F_i} (1 - F_i(t_i)) c_i a_i(\vec{v}, \vec{c}, c_0) f_{-i}(v_{-i})] dv \, dg \, dg^o$$

$$= -\mathbf{E} \sum_{i=1}^{N} [U_i(a, p, l_i)] - \sum_{i=1}^{n_1} \int_G \int_F \int_{G^o} \frac{(1 - F_i(v_i))}{f_i(v_i)} c_i a_i(\vec{v}, \vec{c}, c_0) df \, dg \, dg^o$$

This implies that:

$$I_1 + I_2 = -\mathbf{E}\sum_{i=1}^{n_1} U_i(a,p,l_i) + \alpha\mathbf{E}c_0$$
$$+ \int_G \int_F \int_{G^o} \sum_{i=1}^{n_1} a_i(\vec{v},\vec{c},c_0)[c_i(v_i - \frac{1-F_i(v_i)}{f_i(v_i)}) + \alpha(c_i - c_0)]df\,dg\,dg^o \tag{29}$$

Since $-\mathbf{E}\sum_{i=1}^{n_1} U_i(a,p,l_i) \leq 0$, we can set the allocation and payment rule to satisfies $U_i(a,p,l_i) = 0$. This can be achieved by setting the payment rule not lower than $\min(v_i \mid v_i \in V_i)$. Therefore, maximizing $R_1(a,p) + \alpha R_2(a,p)$ is equivalent to maximizing:

$$\int_G \int_F \int_{G^o} \sum_{i=1}^{n_1} a_i(\vec{v},\vec{c},c_0)[c_i(v_i - \frac{1-F_i(v_i)}{f_i(v_i)}) + \alpha(c_i - c_0)]df\,dg\,dg^o \tag{30}$$

$\square$

**Theorem 3.** *The auction defined in Definition 3 has the following properties.*

- *It satisfies IC and IR for all the items, $\forall i$, $v_i^* \geq p_i(v_i^*, c_i^*) \geq 0$.*

- *It maximizes the multi objective $R_1(a,p) + \alpha R_2(a,p)$, which is the same to Equation 5.*

- *The percentage of ads satisfies $\mathbf{E}\lambda_{ad} = \lambda_0$ if and only if*

$$P_{\vec{v}\sim F, \vec{c}\sim G}(max_{i\in\{Ad\}}c_i\Psi(v_i) + \alpha(c_i - c_0) > 0) = \lambda_0. \tag{31}$$

*Proof.* • According to Definition 3, the cost for per click is

$$p_i(v_i^*, c_i^*) = \max\left[\Psi^{-1}((c_j^*\Psi(v_j^*) + \alpha c_j^* - \alpha c_i^*)/c_i^*), \min(v^* \mid v^* \in V_i)\right] \tag{32}$$

We use $l_i$ to denote $\min(v^* \mid v^* \in V_i)$. Since $i = \mathrm{argmax}\, c_i^*\Psi(v_i^*) + \alpha c_i^*$, we have:

$$(c_j^*\Psi(v_j^*) + \alpha c_j^* - \alpha c_i^*)/c_i^* \leq \Psi(v_i^*) \tag{33}$$

In Definition 3, we assume that all the value distributions are normal. This implies that $\Psi$ and $\Psi^{-1}$ are increasing functions of $v_i^*$, which means that:

$$v_i^* \geq \max(\Psi^{-1}[(c_j^*\Psi(v_j^*) + \alpha c_j^* - \alpha c_i^*)/c_i^*], l_i) \geq l_i \geq 0 \tag{34}$$

Since $v_i^* \geq p_i(v_i^*, c_i^*) \geq 0$, we have $p_i(v_i^*, c_i^*) \equiv 0$ if $i \in \{Org\}$. This implies that the mechanism satisfies IC and IR for all organic items.

In the proof of Theorem 2, we have already guarantee the incentive compatiblity of the unified auction. Here we give a simple proof according to Myerson's Lemma. From the Myerson's Lemma, the mechanism satisfies IC constrains for ad items if the allocation rule $a_i(\vec{v}^*, \vec{c}^*)$ is monotonic with respect to value $\vec{v}_i^*$ and the payment rule $p_i(\vec{v}^*, \vec{c}^*)$ satisfies the threshold condition:

$$p_i(\vec{v}^*, \vec{c}^*) = \mathrm{argmin}_v[c_i^*\Psi(v) + \alpha c_i^* = \max_j c_j^*\Psi(v_j) + \alpha c_j^*] \tag{35}$$

Since all the value distributions are normal, we have $\Psi(v_i^*)$ is increasing function of $v_i^*$, and $c_i^*\Psi(v_i^*) + \alpha c_i^*$ is also increasing function of $v_i^*$. Therefore, the allocation rule $a_i(\vec{v}^*, \vec{c}^*)$ is monotonic with respect to value $\vec{v}_i^*$. According to equation (32), we have that when the bidder bidding truthfully, its payment satisfies equation (35). Therefore, the mechanism satisfies IC and IR for all ad items.

- We denote $c_0 = max\vec{c}^o$, $\vec{F} = (F, F^o)$, $\vec{G} = (G, G^o)$, and $\{Ad\}$ as the set of ads, $\{Org\}$ as the set of organics. Then we have:

$$\int_{\vec{F}} \int_{\vec{G}} \sum_{i=1}^{N} a_i(v_i^*, c_i^*)[c_i^*(v_i^* - \frac{1 - F_i(v_i^*)}{f_i(v_i^*)}) + \alpha c_i^*]d\vec{f}d\vec{g}$$

$$= \mathbf{I}(argmaxc_i^*(v_i^* - \frac{1 - F_i(v_i^*)}{f_i(v_i^*)}) + \alpha c_i^* \in \{Ad\})$$

$$\cdot \int_{\vec{F}} \int_{\vec{G}} \sum_{i=1}^{|\{Ad\}|} a_i(v_i^*, c_i^*)[c_i^*(v_i^* - \frac{1 - F_i(v_i^*)}{f_i(v_i^*)}) + \alpha c_i^*]d\vec{f}d\vec{g}$$

$$+ \mathbf{I}(argmaxc_i^*(v_i^* - \frac{1 - F_i(v_i^*)}{f_i(v_i^*)}) + \alpha c_i^* \in \{Org\}) \int_{\vec{F}} \int_{\vec{G}} \sum_{i=1}^{|\{Org\}|} [a_i(v_i^*, c_i^*)\alpha c_i^*]d\vec{f}d\vec{g}$$

$$= \mathbf{I}(argmaxc_i^*(v_i^* - \frac{1 - F_i(v_i^*)}{f_i(v_i^*)}) + \alpha c_i^* \in \{Ad\})$$

$$\int_{\vec{F}} \int_{\vec{G}} \sum_{i=1}^{|\{Ad\}|} a_i(v_i^*, c_i^*)[c_i^*(v_i^* - \frac{1 - F_i(v_i^*)}{f_i(v_i^*)}) + \alpha c_i^*] - \alpha c_0 d\vec{f}d\vec{g}$$

$$+ \int_{\vec{F}} \int_{\vec{G}} \sum_{i=1}^{|\{Org\}|} [a_i(v_i^*, c_i^*)\alpha c_i^*]d\vec{f}d\vec{g}$$

$$= \int_{G^o} \alpha c_0 dg^o + \mathbf{I}(argmaxc_i^*(v_i^* - \frac{1 - F_i(v_i^*)}{f_i(v_i^*)}) + \alpha c_i^* \in \{Ad\})$$

$$\int_{F} \int_{G} \int_{G^o} \sum_{i=1}^{|\{Ad\}|} a_i(v_i, c_i, c_0) \cdot [c_i(v_i - \frac{1 - F_i(v_i)}{f_i(v_i)}) + \alpha(c_i - c_0)]dfdg$$

$$= \alpha\mathbf{E}c_0 + \int_{G} \int_{F} \int_{G^o} \sum_{i=1}^{N} a_i(v_i, c_i, c_0)[c_i(v_i - \frac{1 - F_i(v_i)}{f_i(v_i)}) + \alpha(c_i - c_0)]dfdgdg^o$$

(36)

This emplies that the mechanism maximizes the multi objective $R_1(a, p) + \alpha R_2(a, p)$, which is the same to Equation 5. According to the proof of Theorem 2, it is necessary to set the payment rule not lower that $min (v_i \mid v_i \in V_i)$. Therefore this Theorem holds.

- Given a single slot, the PAE is equal to the probability of the maximum virtual value comes from ads. Therefore, the percentage of ads satisfies $\mathbf{E}\lambda_{ad} = \lambda_0$ if and only if

$$P_{\vec{v} \sim F, \vec{c} \sim G}(max_{i \in \{Ad\}} c_i \Psi(v_i) + \alpha c_i > \alpha c_0) = \lambda_0. \tag{37}$$

$\square$

**Theorem 4.** *Given the weight $\alpha$ and corresponding $\lambda_0 \in (0, 1)$ which satisfies Definition 3, we have the following conclusion:*

$$max_{a,p}R_1(a, p) + \alpha R_2(a, p) > max_{a,p}(1 - \lambda_0)\alpha R_2^{org} + \lambda_0[R_1^{ad}(a, p) + \alpha R_2^{ad}(a, p)] \tag{38}$$

*where the left part comes from Equation 5 and right part comes from Equation 9.*

*Proof.* According to equation (29), we have:

$$max_{a,p}(1 - \lambda_0)\alpha C_{org} + \lambda_0[R_{ad}(a, p) + \alpha C_{ad}(a, p)]$$

$$= (1 - \lambda_0)\alpha \mathbf{E}c_0 + \lambda_0[-\sum_{i=1}^{n_1} \mathbf{E}U_i(a, p, l_i) + \sum_{i=1}^{n_1} \int_F \int_G a_i(\vec{v}, \vec{c})(c_i(v_i - \frac{1 - F_i(v_i)}{f_i(v_i)}) + \alpha c_i)dfdg]$$

$$= \alpha \mathbf{E}c_0 + \lambda_0[-\mathbf{E}\sum_{i=1}^{n_1} U_i(a, p, l_i) + \int_F \int_G \int_{G^\circ} a_i(\vec{v}, \vec{c})[c_i(v_i - \frac{1 - F_i(v_i)}{f_i(v_i)}) + \alpha c_i] - \alpha c_0 dfdgdg^o]$$

$$\leq \alpha \mathbf{E}c_0 + \lambda_0[-\mathbf{E}\sum_{i=1}^{n_1} U_i(a, p, l_i) + \int_F \int_G \int_{G^\circ} a_i(\vec{v}, \vec{c})[c_i(v_i - \frac{1 - F_i(v_i)}{f_i(v_i)}) + \alpha c_i - \alpha c_0]dfdgdg^o]$$

$$< -\sum_{i=1}^{n_1} U_i(a, p, l_i) + \alpha \mathbf{E}c_0 + \int_G \int_F \int_{G^\circ} \sum_{i=1}^{n_1} a_i(\vec{v}, \vec{c}, c_0)[c_i(v_i - \frac{1 - F_i(v_i)}{f_i(v_i)}) + \alpha(c_i - c_0)]dfdgdg^o$$

$$= max_{a,p}R_1(a, p) + \alpha R_2(a, p)$$

$$(39)$$

The last inequality holds because when $\lambda > 0$, it is obvious that

$$-\mathbf{E}\sum_{i=1}^{n_1} U_i(a, p, l_i) + \int_G \int_F \int_{G^\circ} \sum_{i=1}^{n_1} a_i(\vec{v}, \vec{c}, c_0)[c_i(v_i - \frac{1 - F_i(v_i)}{f_i(v_i)}) + \alpha(c_i - c_0)]dfdgdg^o > 0$$

$$(40)$$

in equation (29). This implies that

$$\lambda_0[-\mathbf{E}\sum_{i=1}^{n_1} U_i(a, p, l_i) + \int_F \int_G \int_{G^\circ} a_i(\vec{v}, \vec{c})[c_i(v_i - \frac{1 - F_i(v_i)}{f_i(v_i)}) + \alpha c_i - \alpha c_0]dfdgdg^o]$$

$$\leq -\mathbf{E}\sum_{i=1}^{n_1} U_i(a, p, l_i) + \int_G \int_F \int_{G^\circ} \sum_{i=1}^{n_1} a_i(\vec{v}, \vec{c}, c_0)[c_i(v_i - \frac{1 - F_i(v_i)}{f_i(v_i)}) + \alpha(c_i - c_0)]dfdgdg^o$$

$$(41)$$

$\square$

**Remark 1**: In order to illustrate that AMMD framework allows for arbitrary extensions for item attributes, we provide an example of the CVR constraint and the corresponding Pareto Equation. We assume that each item (including ads and organics) has an specific conversion rate $z_i$ (which is independent with other attributes), and the average conversion rate (CVR) of all displayed items is required to be no less than $\rho_0$.

Adopting Lagrange multiplier, the constraint $\mathbf{E}(\bar{z}_i \mid i \in \{\text{exposed items}\}) \geq \rho_0$ makes the objective becomes:

$$\max R_1 + \alpha R_2 - M \text{Relu}[\rho_0 - \mathbf{E}(\bar{z}_i \mid i \in \{\text{exposed items}\})] \quad (42)$$

Since the conversion rate is assumed independent with other attributes, it also forms a Pareto curve with $R_1$ and $R_2$. Therefore, the multi-objective optimization can be simplified as:

$$\max R_1 + \alpha R_2 + \sum_{i \in \{\text{all items}\}} \beta z_i \quad \text{s.t.} \ \mathbf{E}(\bar{z}_i \mid i \in \{\text{exposed items}\}) = \rho_0 \quad (43)$$

Similar to the proof of Theorem 3, we can derive the equivalence between maximizing multi-objective revenue and the virtual value function $c_i \Psi(v_i) + \alpha c_i + \beta z_i$. Then we have the following Pareto Equations in independent identical multi-slot auctions (generalized multi-slot auctions can be calculated similarly):

$$P_{\vec{v} \sim F, \vec{c} \sim G, \vec{z} \sim Z}(max_{i \in \{Ad\}}[c_i \Psi(v_i) + \alpha c_i + \beta z_i] - (max_{i \in \{Org\}}[\alpha c_i + \beta z_i]) > 0) = \lambda_0$$
$$\mathbf{E}_{\vec{v} \sim F, \vec{c} \sim G, \vec{z} \sim Z}(z_i \mid i = \text{argmax} \ c_i \Psi(v_i) + \alpha c_i + \beta z_i) = \rho_0$$

$$(44)$$

Given the traffic characteristics, we can solve the weight parameters $(\alpha, \beta)$ according to the above two equations. This implies that AMMD framework allows for extensions in item attributes and objective constraints.

**Remark 3**: The application of VCG pricing rule guarantees the incentive compatibility of AMMD. For a complete proof, please refer to the work of (Varian & Harris, 2014). Here we give a simple proof sketch.

Given an advertisement bidder $i$ and its value $v_i$, click-through rate $c_i$. We assume that if it bids truthfully $b_i = v_i$, it will win the $k$th slot.

- For any untruthful bidding $b_i'$, if it still wins the $k$th slot, then its utility is the same with truthful bidding. This is obvious because the utility of bidder is determined by its allocation and payment. Given the same allocation, the payment of the bidder is independent with its bidding.

- Then we prove that for any untruthful bidding $b_i'$, if it wins a worse slot (taking the $k + 1$th slot as an example), then its utility is the same with truthful bidding.

  We assume that the value of the current bidder is $v_0$, and the CTR of the slot allocated to it is $c_0$. There are $n$ slots worse than this slot, with CTR $c_1, \cdots, c_n$ satisfying $c_0 \geq c_1 \geq \cdots$. These slots are allocated to bidders with $v_1, \cdots, v_n$ values. We use $q_i(v_i)$ to denote the virtual value of the bidder $i$. Here we assume that all the virtual value functions $q_i$ and their inverse functions $q_i^{-1}$ are linear functions (It holds when the value distributions of bidders are uniform distributions).

  Under these assumptions, the payment decrease of the current bidder for untruthful bidding is:

$$q_0^{-1}\Big[\frac{\sum_{j=1}^{n+1} c_{j-1}q(v_j) - \sum_{j=1}^{n} c_j q(v_j)}{c_0}\Big] \cdot c_0 - q_0^{-1}\Big[\frac{\sum_{j=1}^{n} c_{j-1}q(v_j) - \sum_{j=1}^{n} c_j q(v_j)}{c_1}\Big] \cdot c_1$$

$$= q_0^{-1}\Big[\sum_{j=1}^{n+1} c_{j-1}q(v_j) - \sum_{j=1}^{n} c_j q(v_j)\Big] - q_0^{-1}\Big[\sum_{j=1}^{n} c_{j-1}q(v_j) - \sum_{j=1}^{n} c_j q(v_j)\Big]$$

$$= q_0^{-1}[c_0 q_1(v_1) - c_1 q_1(v_1)]$$

$$\leq q_0^{-1}(c_0 q_0(v_0) - c_1 q_0(v_0))$$

$$= v_0(c_0 - c_1).$$

(45)

  This implies that the payment decrease is lower than its value decrease, which means that the utility of untruthful bidding is lower than truthful bidding.

According the above two conditions, we prove that the utility of untruthful bidding is lower than truthful bidding in specific assumptions. Therefore the mechanism is incentive compatibility. We also verified the incentive compatibility of the mechanism in the experiments.

**Remark**: In online controlling process, we adopt the PID controller because of its good theoretical properties and wide application. Adjusting model parameters to control the percentage of advertisement exposed in online traffic scenarios can be viewed as a time-varying signal tracking problem in a nonlinear stochastic system. Theoretical analysis on the ability of the classical PID controller can be referred to in the work of Zhao & Yuan (2024).

Briefly speaking, such control systems can be stabilized in the mean square sense, provided that the three PID gains $(k_p, k_i, k_d)$ are selected from a stability region. The steady-state tracking error has an upper bound proportional to the sum of the varying rates of the reference signals, the varying rates of the disturbances and random noises. In our experiments, we found that the AMMD framework with this online controller met the constraints such as the PAE and CVR.

**Remark**: The definition of Utopia distance is detailed as follows: For points $(x_1, y_1), \cdots, (x_n, y_n)$ and arrays $(x^{\vec{k}1}, y^{\vec{k}1}), \cdots, (x^{\vec{k}j}, y^{\vec{k}j})$, the Utopia point is defined as $(x_0, y_0)$, where

$$x_0 = \max (x_1, \cdots, x_n, x^{\vec{k}1}, \cdots, x^{\vec{k}j}), \ y_0 = \max (y_1, \cdots, y_n, y^{\vec{k}1}, \cdots, y^{\vec{k}j}) \quad (46)$$

The utopia distance of $(x_i, y_i)$ is defined as

$$d(x_i, y_i) = [(x_i - x_0)^2 + (y_i - y_0)^2]^{\frac{1}{2}} \quad (47)$$

The utopia distance of $(x^{\vec{k}s}, y^{\vec{k}s})$ is defined as

$$d(x^{\vec{k}s}, y^{\vec{k}s}) = \min_{x \in x^{\vec{k}s}, y \in y^{\vec{k}s}} [(x - x_0)^2 + (y - y_0)^2]^{\frac{1}{2}} \quad (48)$$

## A.2 Algorithms of AMMD

### A.2.1 Training Hypernetwork and Model Network

For modeled auctions, both the allocation and charging rules are determined by the model parameters. According to the theory in Section 3, the revenue of the incentive-compatible mechanism are only affected by the allocation rules. Since the AMMD framework adopts VCG pricing rule to ensure IC, the multi-objective optimization only needs to be run on the modeled virtual value function. The network consists of two parts, a fixed parameter part $\theta_{-w}$ and a controllable parameter $\theta_w$ part. We hope to obtain a family of mechanisms that are applicable to any weight parameters and traffic characteristics during offline training. Therefore, each training sample contains not only item attributes, but also randomly sampled weight parameters. Detailed algorithm is provided in Algorithm 1.

---

**Algorithm 1** Training Hypernetwork and Model Network

---

**Input**: Hypernetwork $H(\cdot, \theta_H)$, Modeled network $M(\cdot, [\theta_w, \theta_{-w}])$, Multi-objective loss function $L$, Value distribution $F$, CTR and CVR distribution for ads $C, Z$ and organics $C^o, Z^o$, Weight distribution $D_w$.

1: **repeat**
2:     Sample training sample $(\vec{v}, \vec{c}, \vec{z})$ from $(F, C, C^o, Z, Z^o)$ and $\vec{w}$ from $D_w$
3:     Hypernetwork generates parameters $\theta_w = H(\vec{w}, \theta_H)$
4:     Input $(\vec{v}, \vec{c}, \vec{z})$ to $M(\cdot, [\theta_w, \theta_{-w}])$ to derive allocation and payment $(a, p)$
5:     Compute loss $L(a, p, \vec{w}, \vec{v}, \vec{c}, \vec{z})$
6:     Update parameters $\theta_H$ and $\theta_{-w}$ based on loss using back-propagation
7: **until** Maximum number of iterations reached
8: **return** $H(\cdot, \theta_H), M(\cdot, \theta_{-w})$

**Output**: Hypernetwork $H(\cdot, \theta_H)$, Fixed parameters for modeled network $M(\cdot, [\theta_w, \theta_{-w}])$.

---

### A.2.2 AMMD (Adaptive modeled mechanism design)

The AMMD framework does not update model parameters in online applications, but adjusts multi-objective weights based on KPI feedback. Details algorithm is provided in Algorithm 2.

---

**Algorithm 2** AMMD (Adaptive modeled mechanism design)

---

**Input**: Hypernetwork $H(\cdot, \theta_H)$, Modeled network $M(\cdot, [\theta_w, \theta_{-w}])$, initialized parameter $(\alpha, \beta)$, number of episodes $N_1$, number of auctions in one episode $N_2$, k slots with click-through rate $c^1, \cdots c^k$.

1: **for** $i_1$ in range($N_1$) **do**
2:     **for** $i_2$ in range($N_2$) **do**
3:         Receive all item attributes $(\vec{v}_{ad}, \vec{c}_{ad}, \vec{z}_{ad})$ and $(\vec{v}_{org}, \vec{c}_{org}, \vec{z}_{org})$
4:         Generating weight parameters by distribution approximation $\hat{F} = \hat{F}(\vec{v})$, $\hat{C} = \hat{C}(\vec{c})$ and $\hat{Z} = \hat{Z}(\vec{z})$, $w = [(\alpha, \beta), \hat{F}, \hat{C}, \hat{Z}]$
5:         Calculate the virtual value for each item $\theta_w = H(w, \theta_H)$, $v_i^* = M[(v_i, c_i, z_i), (\theta_w, \theta_{-w})]$
6:         Ranking the items with $v^*$ and impressing the top k items, adopting VCG pricing rule in equation (12) for these items
7:         Record the cumulative clicks, costs, and KPI feedbacks
8:     **end for**
9:     Update parameters $(\alpha, \beta)$ with the PID controller
10: **end for**
11: **return** Total clicks and costs

---

### A.3 Additional experiments and experiment details

All experiments in this paper were run with one A100 GPU, 40 GB memory support. Experiments can be performed under both Windows and Linux systems. The neural network module is built using the pytorch framework. For specific details, please refer to the code submitted in the appendix.

#### A.3.1 Incentive compatibility test for mechanisms

In order to show that AMMD satisfies the incentive compatibility, we design an experimental verification. We randomly sample several groups of different item attribute samples and calculate the allocation and payment under the trained model. In each auction, we randomly select an ad item and set its bid to be untruthful ($b = 0.99v, 0.95v.0.9v, 0.8v$). By comparing the change in the advertiser's average utility compared with the truthful bidding, we analyze whether the mechanism meets incentive compatibility. The experimental setting is the same to generalized multi-slot auctions without CVR constraint in dynamic environments (setting is detailed in the following section).

The experimental results are given in Figure 6. From the figure, we can see that, except for GSP, the utility of all mechanisms under untruthful bids is less than that under truthful bids. This result shows from an experimental perspective that AMMD satisfies incentive compatibility. The experimental results also show that the outcome of the GSP mechanism in actual application is significantly lower than that in the experiment of this paper. This is because the failure to meet incentive compatibility will cause advertisers to lower their bids. As can be seen from Figure 6, the actual outcome of GSP is at least reduced by 20%.

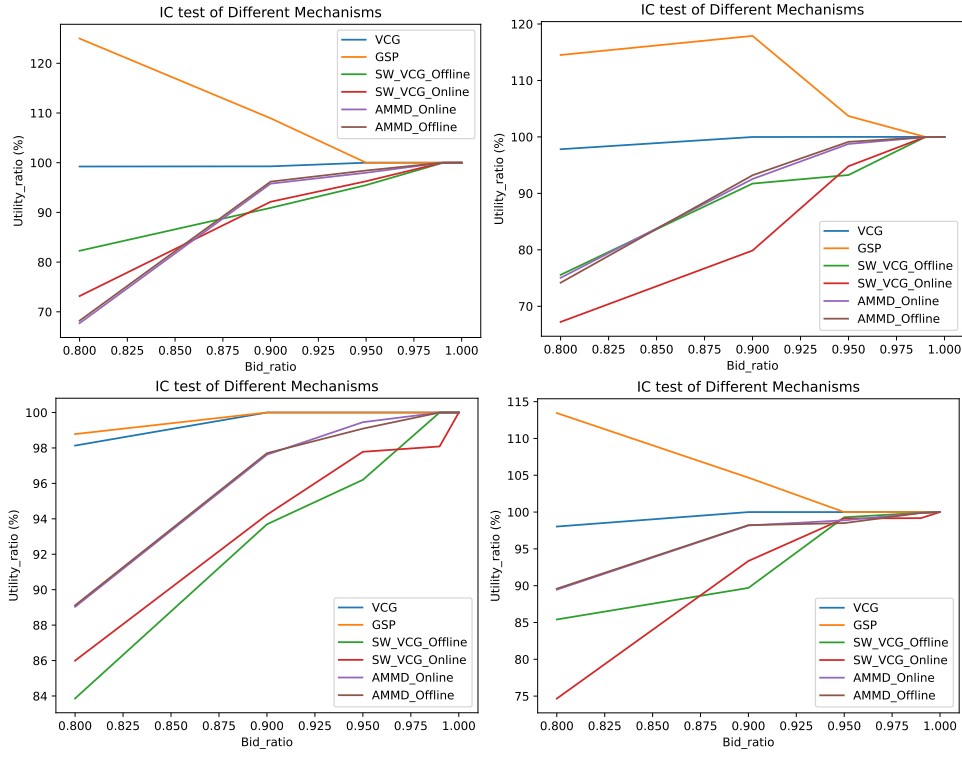

Figure 6: Experiments for testing utility change when bidder bidding untruthfully.

#### A.3.2 Experiment details in independent identical multi-slot auctions

In independent identical multi-slot auctions, we first conduct experiments in static environments. The existence of the replaced layer in AMMD allows it to switch to the corresponding mechanism in the optimal mechanism family according to the distribution characteristics. In all experiments, 1000 auctions are conducted per time unit and the results are averaged as feedback for network parameter updates.

|  | Click | | Cost | | 100%-PAE | |
|---|---|---|---|---|---|---|
|  | RegretNet | AMMD | RegretNet | AMMD | RegretNet | AMMD |
| F~U[0,0.5] | 0.805 | 0.817 | 0.018 | 0.084 | 49.6% ± 2.8% | 50.5% ± 3.2% |
| F~U[0,1.0] | 0.779 | 0.815 | 0.061 | 0.177 | 49.8% ± 2.3% | 50.3% ± 3.0% |
| F~U[0,1.5] | 0.764 | 0.807 | 0.129 | 0.265 | 50.3% ± 2.1% | 49.6% ± 3.4% |
| F~U[0,2.0] | 0.748 | 0.802 | 0.193 | 0.380 | 49.8% ± 2.6% | 48.7% ± 2.9% |

Table 2: AMMD Mechanism application in different traffic with hypernetwork.

We compare the AMMD model trained with random samples with RegretNet in different traffic distributions. For the AMMD algorithm, all items are ranked by a unified virtual value network. When the exposure ratio of advertisements and organic items does not meet the PAE constraint, the controller will adjust the network parameters according to the feedback until it converges to a state that meets the constraint. However, for RegretNet, it can only be trained using a certain loss function. Here we give the comparison results of using $Costs + \alpha Clicks, \alpha = 0.3, 0.5$ as loss functions. The PID control module of AMMD is initialized with $\alpha$=0.5, and its $(k_p, k_i, k_d)$ is selected as (0.005, 0.0002, 1), with $l = 24$. In other subsequent experiments, the selection of this hyperparameter is similar. The specific details can be found in the code in the supplementary material.

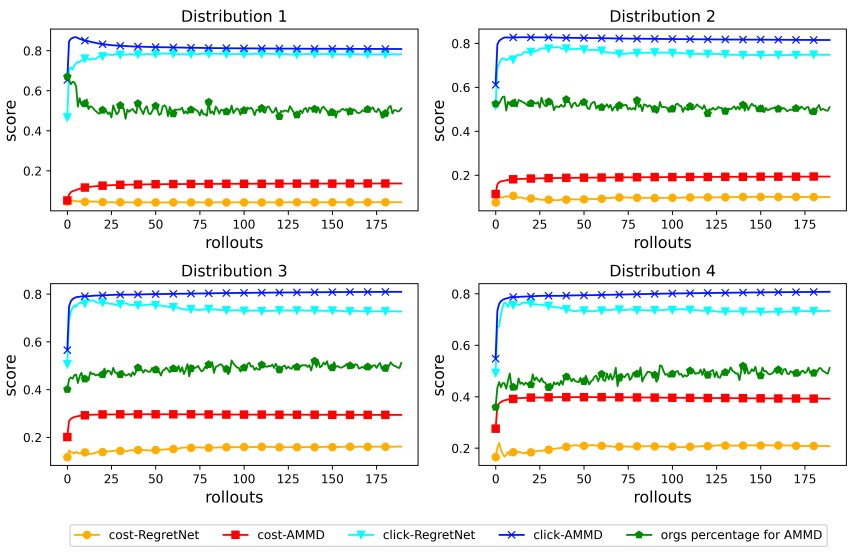

Figure 7: $\alpha = 0.3$

Detailed results of Figure 8 is given in Table 2. The reason why RegretNet does not strictly meet the PAE constraint is that there are 1,000 auctions in each batch. Even if the probability of allocating ads or organics is 50%, there will still be a certain error. This experimental result shows that a well-trained AMMD framework can output the optimal auction mechanism under any traffic distribution and surpass existing methods by combining multi-objective optimization with online control.

The comparison between AMMD and the Pareto curve of RegretNet in dynamic environments is presented in Figure 4. Here, we provide detailed results from this experiment. In this setup, the click-through rates (CTR) of all items are derived from independent static distributions, while the upper bound of the value distribution changes periodically over time. Specifically, the value distribution of the advertising items is uniformly distributed with a lower bound of 0, and its upper bound varies evenly over time within a range of 0.5 to 1.5, following a 24-rollout cycle. This simulates the willingness of advertisers to pay during different time periods on the e-commerce platform.

From Table 3, we observe that when there are no additional constraints, the adjustment of multi-objective weights strictly adheres to the definition of the Pareto region; it is impossible to improve multiple objective functions simultaneously by changing the weights. When constraints such as the PAE are introduced, these constraints can be viewed as a plane subset in the overall space.

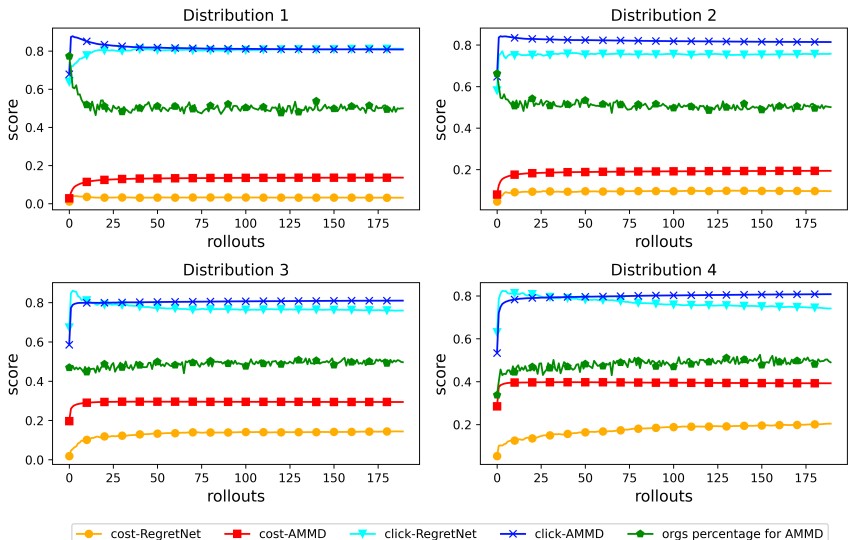

Figure 8: $\alpha = 0.5$

| Mechanisms | $\alpha$ | click | cost |
|---|---|---|---|
| offline RegretNet | 0.1 | 0.683 | 0.103 |
| | 0.5 | 0.775 | 0.074 |
| | 1.2 | 0.808 | 0.039 |
| | 2.0 | 0.815 | 0.025 |
| online RegretNet | 0.1 | 0.687 | 0.104 |
| | 0.5 | 0.772 | 0.076 |
| | 1.2 | 0.811 | 0.037 |
| | 2.0 | 0.815 | 0.028 |
| offline AMMD | adaptive | 0.836 | 0.177 |
| online AMMD | adaptive | 0.839 | 0.184 |

Table 3: Comparison between AMMD and Pareto curve of RegretNet.

The intersection of this subset with the Pareto region represents the points that satisfy the Pareto equation. Achieving multi-objective Pareto optimality under constraints requires solving the Pareto equation to determine the optimal weights. This is the key reason why AMMD can outperform existing methods.

### A.3.3 EXPERIMENTS IN GENERALIZED MULTI-SLOT AUCTIONS

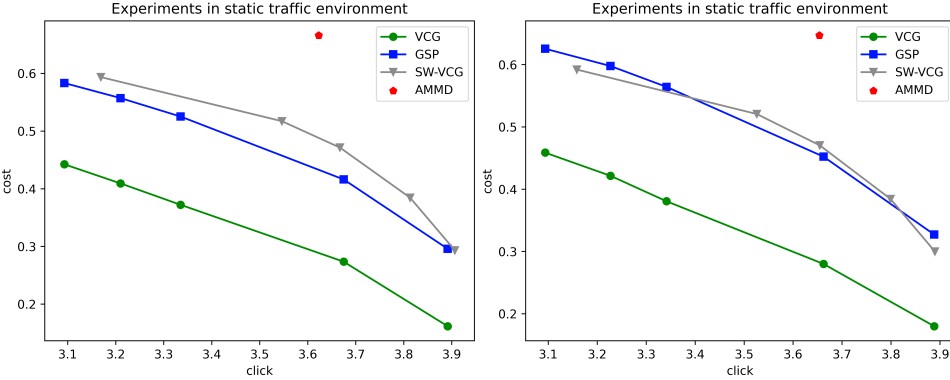

Figure 9: Generalized multi-slot auctions in static setting

In this section we present the experimental setup and results for generalized multi-slot auctions in detail. In all scenarios, four ad items and four organic items compete for four advertising slots. The click-through rates of the four advertising slots are set to (1.0,0.8,0.6,0.5), and the click-through rates of advertising items are sampled from uniform distribution $U[0, 1]$, and the click-through rates of organic items are sampled from uniform distribution $U[0, 2.5]$. When there are CVR constraints, the CVR of each item is sampled from uniform distribution $U[0, 0.175]$ with the constraint $\bar{z} \geq 0.1$.

In static scenarios, the value distribution of ad items are sampled from static uniform distribution $U[0, 1.5]$. For VCG and GSP mechanisms, all items are sorted according to the metric $Value \times CTR + \alpha \times CTR$, where $\alpha$ takes $(0.01, 0.1, 0.2, 0.5, 1.0)$. In the uniformly sorted sequence, the two highest ranked ad and organic items are each selected for impression. In the pricing process for ad items using VCG and GSP rules, items added in suboptimal sequences are restricted to ads to maintain the PAE constraint.

For SW-VCG, it maps the value and CTR of items to virtual values through neural networks and uses VCG rules for pricing. The neural network is trained using the clicks and revenue after sorting and pricing as the loss function $Costs + \alpha Clicks$, where $\alpha$ takes $(0.01, 0.1, 0.2, 0.5, 1.0)$.

In Figure 9 we give the auction results for the different mechanisms under two sets of random sampling. Notice that the GSP mechanism does not satisfy incentive compatibility, and its actual results should be approximately equivalent to the VCG mechanism. From the comparison of the SW-VCG and VCG mechanisms we can see that SW-VCG strictly Pareto dominates the VCG mechanism at different parameter settings. This illustrates that the virtual values obtained using multi-objective training achieve a more optimal allocation compared to the real values.

In the comparison between AMMD and SW-VCG, we can see that AMMD significantly outperforms the Pareto curve formed by the SW-VCG mechanism. Theorem 4 shows that charging the ad items within the unified sequence strictly leads to an increase in total costs. However, for SW-VCG trained with static multi-objective weights, it cannot naturally satisfy the PAE constraint. The process of ensuring that sequences satisfy the PAE constraint (limiting the proportion of ad items in a uniform sequence) makes it infeasible to charge for ads in a uniform sequence. This is due to the fact that suboptimal sequences in VCG pricing are also required to satisfy the PAE constraint. However, for AMMD, the satisfaction of PAE is achieved by adjusting the weights of multiple objectives. This allows ad items to be charged in a uniform sequence, which significantly improves the multi-objective revenue.

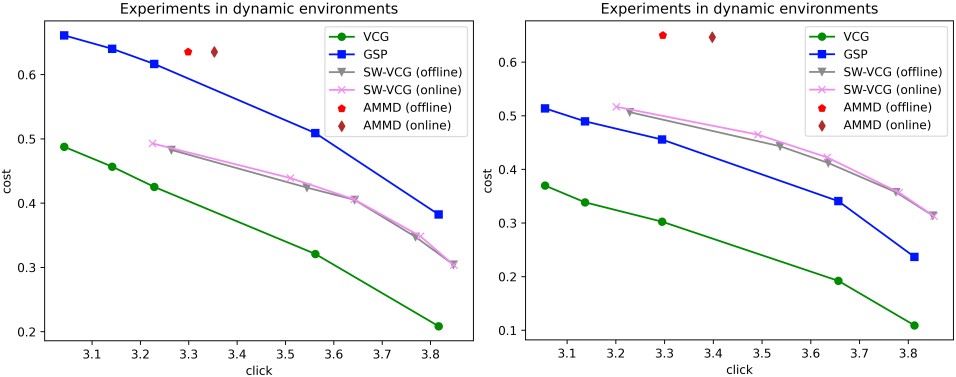

Figure 10: Generalized multi-slot auctions in dynamic setting

In dynamic scenarios, the parameter settings are basically the same as in a static scene. The value distribution of the ad items is the same as the settings in the independent identical auction experiment. The experiment results is given in Figure 10, 11.

In CVR constrained experiment, the selection of $beta$ for GSP, VCG and SW-VCG is from [0.6,2.0,3.0,5.5,8.0]. Other hyper parameters are the same to experiment without CVR constraint. Detailed experiment can been seen in the submitted code.

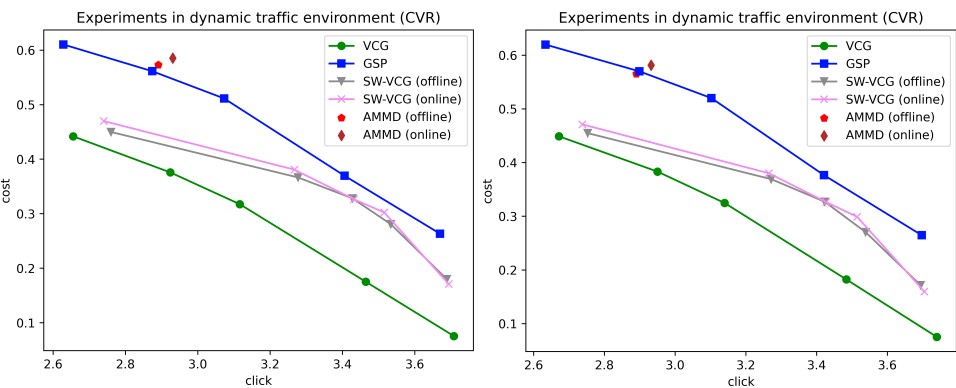

Figure 11: Generalized multi-slot auctions in dynamic setting (with CVR constraint)

