# OpenReview forum: "Online Auction for Ads and Organics"
_ICLR.cc/2025/Conference — ICLR 2025 Conference Withdrawn Submission_

### Official Review · Reviewer_e4u1 · 2024-10-27

**Soundness:** 1
**Presentation:** 2
**Contribution:** 2
**Rating:** 3
**Confidence:** 2

**Summary:**

The authors propose a parametrizable mechanism for mixing ads with organic content in online platofrms. They introduce a unifying definition that can capture both organic items (native content) and ad items, and derive an auction with a parametrizable objective that can weigh between placing more weight in ads or native content.
They define a strategyproof mechanism for any weight parametrization, alongside an online-learning algorithm for finding the optimal weights for the platform's objective.

**Strengths:**

The paper tackles an interesting problem.
In practice, there can be indeed a strong distributional shift in bidder's valuations from day to day (and in the quality of the native user content), meaning that there is room for such a dynamic mechanism. I think the premise of the paper is very interesting.
The authors place the paper well alongside related literature.

**Weaknesses:**

I think sections 2 and 3 leave a lot of room for improvement. Let me highlight some points.
1) "The goal of MD is to achieve Pareto optimality, which involves maximizing platform revenue, advertiser utility, and user experience": Mechanism design can have many different goals, the most common being revenue or utilitarian social welfare. Pareto optimality is a property, not an objective function to be maximized. In fact, it is quite a weak property and almost never the goal.
2) Theorem 1: It sounds like this was something that the authors introduce, as it is not in the preliminaries section, and they even have a proof for it. This is a very well-known result in Optimization. A common citation for this result is: Miettinen, K. (1999). Nonlinear Multiobjective Optimization. Springer.
3) On a minor note, I am not sure if this is Theorem 1 is good motivation to focus on a linear objective. Although intuitively, i do agree with that sentiment.
4) "AMMD ensures incentive compatibility through the VCG payment rules": The VCG payment rule is only incentive compatible under specific constraints (convergence to optimality, optimization objective is social welfare = sum of value of all participants).
5) We decompose he optimal mechanism in a static setting into two components: ... and VCG pricing rule which is proven to satisfy incentive compatibility: Similar note as above.
6) "we focus on constrained multi-objective optimal mechanisms and therefore assume the absence of externalities": I am not sure you can do this in your model. You assume the existance of positions/slots, with decreasing click-through-rates. When the ad of agent i gets placed in position j, it "pushes" all other auctions one position down. There are always externalities, even in your model. All auctions for this problem that i am aware of model externalities. A good reference for this
7) "To avoid this problem, we adopt a payment rule similar to Definition 3": If my understanding is correct, you shift the calculated (Myerson) payments by a value that *is not* a constant, but depends on the agent's reports. If that is the case, there is a very high chance this breaks incentive compatibility. In any case, it needs to be proven that the mechanism is incentive compatible given this payment offset.

**Questions:**

In table 1, it is hard to see which mechanism does better. It would be interesting if you could gray out the winners according to 95% CIs.
How do you define PAE for this experiment? Why is it that only AMMD has a confidence interval attached to its PAE, and how do you measue it?

---

### Official Review · Reviewer_HBXi · 2024-11-02

**Soundness:** 2
**Presentation:** 2
**Contribution:** 1
**Rating:** 3
**Confidence:** 4

**Summary:**

This paper studies the online blending auction mechanism design for ads and organic items. The proposed method ensures Pareto optimality for platform revenue, advertiser utilities and user interest. The auction is built upon Myerson auction by unifying organics and ad items as a traffic item. And the multiple objectives are optimized with a scalarization function. Moreover, a neural network is proposed to model multiple objective weights in a dynamic environment and a pid controller is proposed to ensure a proper proportion of organic items (and ad items). Experiments are conducted on synthetic data to verify the effectiveness.

**Strengths:**

The problem of blending auction mechanism design is interesting and important in industrial applications. The proposed method is simple and the scalarization method for multiple objective optimization is widely used.

**Weaknesses:**

The contributions of this paper are not clear. The contributions on multi-obj optimization and auction theory are rather limited. The auction mechanism proposed is quite straight-forward after unifying the organic and ad items.
For the aspect of multiple objective optimization, the pareto optimality is not a difficult part (considering that scalarization is widely used). The focus of many related studies is to find a pareto front of the mul-obj problems and how to find them in an efficient way. And usually the experiments should reflect that the pareto front discovered by a method “dominate” the front of another method (which is not fully presented in Figure 5).
For the aspect of auction theory, the contribution of auction theory is rather limited. The design of the auction is quite straight-forward if both organic and ad items are unified, which limits its novelty and contributions.
About the efficiency of the proposed modeled auction, the training of the hypernetwork depends a random sampling of weight parameters (\alpha and \beta) in each iteration. This can be computation-expensive if the training data is large and the problem can be even worse when the number of objectives is more (considering the combinations of the objective weights).
About the experiments, the strength of SW-VCG is the design of multi-slot ad auctions under externality influence. As the paper indicates the externality is not considered in this setting, the comparison can be unfair.
The notations are not fully explained: line 149 the notation of p_i^j(a,p,.) is confusing.

**Questions:**

Q1: a pid controller is proposed to adjust the weights for multiple objectives. But does it have conflict with the welfare maximization in a long period of time. Considering that pid controller needs to keep some metric in a certain level, it is possible that the adjusted weights are sub-optimal in a long period of time.

Q2: what are the most significant contributions and novelties of this paper?

---

### Official Review · Reviewer_8QXA · 2024-11-03

**Soundness:** 2
**Presentation:** 2
**Contribution:** 2
**Rating:** 3
**Confidence:** 4

**Summary:**

This paper considers an online blending auction design for ads and organics to achieve Pareto optimality for multiple objectives, specifically focusing on maximizing revenue while satisfying ad exposure and CVR constraints. Theoretical characterization of the pareto optimal mechanisms is given. A modeled mechanism called AMMD is proposed, which consists of a network modeling a score function and a hyper network which adjusts the score function according to different weight parameters, and a PID controller controls the weight parameters in the online process to fulfill the constraints.

**Strengths:**

1. The paper proposes a modeled mechanism with a hyper network structure, allowing to integrate PID control with neural-network-modeled mechanisms, to satisfy realistic constraints.
2. Theoretical characterization provides the motivation for modeling a score function.
3. Extensive experiments are conducted, where various baselines are implemented and compared in different settings.
4. The experimental results show that the proposed model effectively satisfies the constraints in dynamic environment and improves the Pareto curve.

**Weaknesses:**

1. The "VCG" pricing rule in (12) is generally not incentive compatible when the modeled virtual value function is non-linear and there are multiple slots. This can be a crucial fault and may affect the soundness of the experimental results in multi-slot settings. A correct payment rule derived from Myerson's lemma can be found in the SW-VCG paper.

2. The theoretical characterization result in section 3 does not appear to be a very novel contribution, as it is basically applying Myerson's lemma in a standard manner to convert the revenue in the objective into virtual values. This approach is also found in prior works such as Bachrach et al EC'14, [Li et al, 2022] and [Li et al, 2024a]. Moreover, this section only discusses the single slot setting, leading to limited generality. Expanding the analysis to multiple slots does not seem technically challenging. Also, the methodology to unify the ads and organics has appeared in existing works such as [Li et al, 2022] and [Li et al, 2024a], so the abstract seems slightly overclaiming or misleading.

3. Although the AMMD model in section 4 gains motivation to model the virtual value function from definition 3 and theorem 3, the structural results are not fully exploited. As mentioned just after theorem 3, to control the proportion of ads, it suffices to control the $\alpha$. Therefore, a natural approach is to learn Myerson's virtual value function $\Psi_i(v_i)$ for each ad independent of $\alpha,\beta$, and calculate the scores as the form in definition 3 (similar to the structure of SW-VCG), while using PID to control $\alpha,\beta$ to satisfy the constraints. Consequenltly, the necessity of the purposed model structure needs to be justified by an ablation study.

4. The description of the model structure and experimental setting is sometimes unclear. In section 4, the network structure of model $M$ and how to calculate the inverse function $q^{-1}$ of the neural network are not provided. In section 5.1, the claim "Since the auctions between different slots are independent, this setting can be simplified to repeated single-slot auctions" lacks support as the "independent identical multi-slot auction scenario" is not formally defined. It may be clearer to just say that the single slot scenario is considered. In section 5.2, more explanations are needed on the difference between online and offline settings and the implication of Utopia distance.

5. The implementation of the baseline SW-VCG is questionable. Theoretically, SW-VCG will achieve the optimal Pareto curve assuming that the virtual value functions are learned perfectly. As the virtual value function of uniform distribution is just a linear function, I think the outcome of SW-VCG should not be too far from the optimal curve.

6. Minor issues:
P5, a distribution with increasing virtual value is usually called a regular distribution rather than a normal distribution.
P6, the formula (11) is not presented very rigorously.
P9, the citation for SW-VCG should be [Li et al, 2023].

**Questions:**

1. Please explain concretely how the ads and organics are processed in the calculation of the allocation and payment of the baseline auctions.
2. Can the proposed hyper network structure be applied in more complicated scenarios where the approach I mentioned in Weaknesses.3. is not applicable?

---

### Official Review · Reviewer_XKDJ · 2024-11-03

**Soundness:** 2
**Presentation:** 2
**Contribution:** 1
**Rating:** 3
**Confidence:** 3

**Summary:**

This paper studies the blending auction problem and proposes a mechanism called AMMD, which trains a neural network to calculate the virtual value of each item and can dynamically update the mechanism parameters in an online service environment. Extensive experiments demonstrate that AMMD outperforms existing methods in both click-through rates and revenue.

**Strengths:**

1. The hypernetwork module can adapt the mechanism parameters to the online environment with changing characteristics.

2. According to the experimental results, the proposed AMMD method seems to be able to achieve relatively higher revenue than existing IC methods.

**Weaknesses:**

1. This paper is not novel enough. Just giving a unified name of organic items and ad items is not innovative, let alone a contribution. Most previous work on blending auctions has implicitly unified them. Most theoretical results in section 3 come from existing work; the idea of ​​training networks to output virtual values and then using the VCG prcing rule to ensure the incentive-compatibility constraints i not a new idea; using the pid controller to ensure the proportion of organic results is also not new.

2. I don't quite understand why this paper emphasizes the concept of Pareto. If the two objectives have been added with the weight alpha, then the optimization objective is one-dimensional and has nothing to do with Pareto. In fact, I think the modeling is problematic. When modeling the problem, if one wants to optimize the weighted sum of two objectives, the weight coefficient alpha should be a fixed value, but according to section 4.3, alpha is dynamically updated via a PID controller. Note that the PAE constraint in definition 1 is not discussed at all in the analysis of section 3. Therefore, I am wondering whether the authors are actually optimizing the following problem:

      max R_1

      s.t. R_2 >= lambda

    In this way, the dual function is R_1 + alpha(R_2 - lambda) which is consistent with the weighted sum R_1 + alpha R_2 that this paper is trying to optimize. The PAE constraint leads to the dual variable alpha in the optimization objective, rather than alpha being originally in the objective.

3. Many critical things are put into the appendix, including the pseudo code of the algorithm, the training details and the definition of metrics.

**Questions:**

I hope the authors can provide some intuitions on why AMMD proposed in this paper can perform better than SW-VCG, since both methods use data-trained networks to calculate virtual values ​​and use VCG pricing rule afterwards.

---

### Note · Authors · 2024-11-18

I have read and agree with the venue's withdrawal policy on behalf of myself and my co-authors.